# A Sustainable Strategy for Marker-Assisted Selection (MAS) Applied in Grapevine (*Vitis* spp.) Breeding for Resistance to Downy (*Plasmopara Viticola*) and Powdery (*Erysiphe Necator*) Mildews

**DOI:** 10.3390/plants13142001

**Published:** 2024-07-22

**Authors:** Tyrone Possamai, Leonardo Scota, Riccardo Velasco, Daniele Migliaro

**Affiliations:** CREA—Research Center for Viticulture and Enology, 31015 Conegliano, Italy; scotaleonardo1998@gmail.com (L.S.); riccardo.velasco@crea.gov.it (R.V.)

**Keywords:** crude DNA, SSR markers, multiplex PCR, grapevine resistance, *Rpv*, *Ren*, *Run* loci

## Abstract

Plant breeders utilize marker-assisted selection (MAS) to identify favorable or unfavorable alleles in seedlings early. In this task, they need methods that provide maximum information with minimal input of time and economic resources. Grape breeding aimed at producing cultivars resistant to pathogens employs several resistance loci (*Rpv*, *Ren*, and *Run*) that are ideal for implementing MAS. In this work, a sustainable MAS protocol was developed based on non-purified DNA (crude), multiplex PCR of SSR markers, and capillary electrophoresis, and its application on grapevine seedlings to follow some main resistance loci was described. The optimized protocol was utilized on 8440 samples and showed high efficiency, reasonable throughput (2–3.2 min sample), easy handling, flexibility, and tolerable costs (reduced by at least 3.5 times compared to a standard protocol). The *Rpv*, *Ren*, and *Run* allelic data analysis did not show limitations to loci combination and pyramiding, but segregation distortions were frequent and displayed both low (undesired) and high rates of inheritance. The protocol and results presented are useful tools for grape breeders and beyond, and they can address sustainable changes in MAS. Several progenies generated have valuable pyramided resistance and will be the subject of new studies and implementation in the breeding program.

## 1. Introduction

Numerous DNA extraction protocols are currently available for plant tissues and some of them are also suitable for plant species, such as grapevines, which are rich in secondary metabolites that interact with nucleic acids and in enzymatic reactions [1]. Commercial DNA extraction kits, based on 96 well plates, increase the process throughput, but their application remains expensive for larger numbers of samples. On the other hand, cheaper extraction protocols usually require many tedious manual steps (e.g., several centrifugations and washings) and rely on toxic organic solvents, and their adaptation to high throughput methods is rather difficult. Enzymes and reagents for the analysis of non-purified DNA (crude) or direct PCR on plant tissues represent interesting alternatives to DNA extraction [2,3,4,5].

Microsatellites or simple sequence repeats (SSRs) molecular markers are widely used in plant genetic research, for example, in accessions fingerprinting (e.g., Testolin et al. [6]), because of their high levels of polymorphism, reliability, and technical simplicity in their analysis. SSRs were first genotyped on gel electrophoresis, but such a technique was laborious and had a low throughput because it depended on manual labor. This bottleneck was broken by employing capillary electrophoresis and automated allele scoring of fragments labeled with fluorescence dyes, which remain the major technologies used in SSR genotyping [7]. This solution provides precise genotypic data with higher throughput but has increasing costs for the reagents for genotyping.

Plant breeders typically need to screen large numbers of plants. Marker-assisted selection (MAS) solutions enable the detection of favorable alleles in the early stages of development and allow a significant time reduction in breeding evaluations [8]. This technique also enables the detection of phenomena affecting allele frequencies, such as outcrossing, selfing, and/or low viability alleles. Therefore, it is essential to develop MAS methods that deliver maximum information with minimum input of time and economic resources. From a technical perspective, DNA extraction procedures may take most of the costs and time. For this reason, multiplex PCR of SSR coupled with crude DNA amplification, or direct PCR, appeared as an interesting solution to speed up and reduce the cost of genotyping for MAS purposes. SSR-based MAS from non-purified DNA in 96 plates was only proposed for *Oryza sativa* [9].

Grapevines are one of the most widely grown fruit crops. Most cultivated grape varieties belong to the species *Vitis vinifera* L. and are susceptible to downy and powdery mildews [10,11,12] caused by *Plasmopara viticola* and *Erysiphe necator*, respectively. Mildew disease management programs rely on the extensive use of agrochemicals [13], which has negative socioeconomic and environmental impacts [14,15,16]. Alternatively, cultivars resistant to pathogens are a more sustainable solution for viticulture [17]. Over the past twenty years, mapping studies identified more than 30 loci of resistance to *P. viticola* (*Rpv*) and 15 loci of resistance to *E. necator* (*Ren* or *Run*) (e.g., [18]; www.vivc.de), which are available for implementing molecular selection and pyramiding [19]. Some of the main loci used in breeding programs are *Rpv1* [20], *Rpv3* haplotypes [21,22,23], *Rpv10* [24], *Rpv12* [25], *Run1* [26], *Ren1* [27], *Ren3* [28], and *Ren9* [29].

Since the beginning of the grapevine breeding program at CREA—Research Centre for Viticulture and Enology (CREA-VE), different strategies have been utilized for the MAS of seedlings. Primarily, the method proposed by Culley et al. [30], based on ‘tailed primers’ (forward-tailed primer, labeled tail primer, and reverse primer [31]), was used for exploring the SSR and alleles associated with resistance loci to develop an optimal combination of SSR and select the first resistance plant of the project [32]. The introduction of new loci and the increasing number of seedlings justified the adoption of forward-labeled primers that simplify and accelerate multiplex PCR optimization and analysis [33]. However, with several thousands of seedlings per year to analyze, costs and time for DNA extraction became the main bottleneck in MAS. Between 2019 and 2020, we optimized a DNA extraction-free protocol for the screening of large populations which had high efficiency, reasonably low effort and throughput, easy handling, and tolerable costs. In this work, we present this MAS procedure applied for the screening of *Rpv*, *Ren*, and *Run* resistance loci in grapevine populations and the results that emerged from three years of analysis (2020, 2021, and 2022).

## 2. Results

### 2.1. MAS Protocol and Genotyping

The optimized protocol for the MAS of grapevine populations was finally based on the following: home-made buffers (one lysis and one precipitation buffer with no toxic substances), a simple procedure to isolate the crude DNA extract from the leaf, a Taq polymerase for direct-PCR adapted to plant tissues, multiplex PCR of SSR, and pooled amplicons before capillary electrophoresis.

Between 2020 and 2022, the protocol was applied to a total of 8440 grapevine seedlings belonging to 9 cross populations (130–1620 individuals) for screening the presence of *Rpv*, *Ren*, and *Run* loci (Table 1 and Appendix A). In the different years, after sampling and lyophilization of leaf tissue, about 1000 samples per week were processed within a month. In the following analysis, crude DNA was suitable for further amplifications for up to four months.

The different multiplex PCR (one duplex, two triplex, and four four-plex; Appendix A) were optimized by using crude DNA extracts starting from information previously collected in standard PCR conditions [33]. The crude-DNA-based PCR of progenies provided alleles, PCR amplicons yields and uniformity consistent with those of parental plants characterized by using extracted DNA and the standard PCR conditions (Table 1 and Appendix A; [33]). Fluorescence intensities assessed by the ABI Prism 3130xl (Thermo Fisher Scientific, Waltham, MA, USA) DNA analyzer fell within a range optimal for allele sizing (i.e., 1000–8000 relative fluorescence units (RFU)) for about 73% of reactions; about 19% had low fluorescence intensity (<1000 rfu), and 8% had off-scale fluorescence intensity (>8000 rfu). Particularly, markers Gf15-28 and Gf15-30 showed low fluorescence intensity for about 35% and 74% of their alleles, respectively, which represented about 8% of the low fluorescence intensity signals (Figure 1).

A GeneMapper analysis provided all the allelic data for about 98.0% of plant samples, partial data for about 1.5% of samples, and only 0.5% of samples with no fluorescence signals. According to the population, only between 0.0% and 3.5% of seedlings were amplified twice because of relevant missing data (fluorescence intensities/allelic data were not sufficient to clarify if the resistance loci were inherited). The second amplification with better performing conditions (see Section 4) provided complete data for about 50% and 100% of reanalyzed samples suggesting both experimental errors in the previous PCR and the presence of some “low-quality” crude DNA. Finally, complete seedlings allelic profiles ranged between 92.7% and 99.8%, partial profiles between 0.0% and 6.3%, and ‘blank’ profiles between 0.0% and 1.6% (Appendix A). Partial amplification appeared mostly associated with the lack of amplification of the Gf15-28 marker, which provided up to tenfold more missing data (about 2.5% on the overall samples), in comparison to other SSR markers. According to the parental plant’s allelic profiles, ‘null’ alleles were not expected (Appendix A).

The optimized MAS protocol reduced the costs of the material/reagents by at least 3.5 times (i.e., 1.20 € vs. 4.20 € per sample), compared to our previous standard method based on DNA extraction kits, multiplex PCR with standard Taq polymerase, and single capillary electrophoresis runs. The major budget reduction was obtained by substituting the extraction kit (about 85% of the cost of the original analysis) with home-made buffer and Taq for direct-PCR despite the new Taq being up to 3.5 times more expensive than traditional Taq polymerase (i.e., 0.42 € vs. 0.12 € per sample). Moreover, in the new protocol, the materials for sampling and obtaining the crude DNA accounted for about 24% of the total cost (16% is represented by the Collection Microtubes), the PCR reagents (final volume of 20 µL) for about 41% (36% is represented by the direct Taq polymerase), and the capillary electrophoresis reagents for 35% (Figure 2). Samples analyses were even cheaper (with different cost redistribution) for samples amplified in the 12.5 µL final volume PCR (about 25% of samples), pooled before capillary electrophoresis (about 66% of samples), and analyzed in 2020 and 2021 before the doubled Taq polymerase price (minimum cost of the materials of the analysis of about 0.55 € per sample).

Lastly, we estimated a hands-on average time per plant sample of between 2.5 and 3.2 min for the phases that encompass the preparation and organization of the MAS and reagents, sampling (about 40–50 min per 192 samples), lyophilization management (load and unload of the instrument), crude DNA extraction (about 150–180 min per 192 samples), PCR preparation (about 40–50 min per 192 samples), capillary electrophoresis set up (about 40–50 min per 192 samples), data analysis, and samples second amplification when necessary.

### 2.2. Grapevine Resistance Loci Segregation

The SSRs for the screening of *Rpv*, *Ren*, and *Run* were chosen for each population according to the inheritable loci (Appendix A), the allelic profile of parental plants (Appendix A), primers fluorescence (for compatibility in the capillary electrophoresis), and the performance of the markers in previous PCR [33]. Allelic data were used to identify the inherited resistance loci in grapevine seedlings and study their segregation. Plants originated from selfing and not belonging to cross, which ranged between 0.5% and 7.0% (2.0% of the overall plants), were discarded (all seed parents were accurately emasculated except ‘Sk-00-1/7’ and ‘Shavtsitska’ because they had female flowers). Seedlings with partial data (2.0%) were not considered in the evaluation of the loci segregation.

In loci segregation, the couple *Rpv1*/*Run1* was considered as a single locus in chromosome 12, as well as the couple *Ren3*/*Ren9* in chromosome 15. The loci couples were mapped close on the grape reference genome ‘PN40024’ [34] and are usually inherited together. For the allele associated with the *Rpv*, *Ren*, and *Run* loci we expected, for each single locus, a regular Mendelian segregation with a ratio of 1:1 (resistance/not resistance alleles) because the markers/loci are located in different chromosomes and are inherited from one parental plant. The only exceptions were the *Rpv3.1* locus in population 20_38d, which should be inherited by 100% of the individuals (‘Vc531.039’ resistance donor is homozygous for *Rpv3.1* resistance allele), and the *Ren3*/*Ren9* in population 21_45, which should be inherited by 75% of the individuals (ratio 1:2:1 for homozygous for *Ren3*/*Ren9* resistance/heterozygous for *Ren3*/*Ren9* resistance/no *Ren3*/*Ren9* resistance alleles). According to these assumptions, possible loci combinations and percentages of inheritance were defined for each population (Appendix A).

The resistance loci showed different segregation patterns in the populations of the study: most of them segregated according to the expected Mendelian proportion, but only the crosses 19_35 (‘Glera’ × ‘Floreal’) and 21_44 (‘Shavtsitska’ × ‘14_05d_023’) showed the expected ratios at all loci. We observed distorted segregations according to a chi-square test for the *Rpv12* and *Rpv1*/*Run1* in 7 of the 9 populations of the study (Table 2 and Appendix A). The data did not evidence distorted segregations for alleles not associated with resistance loci.

*Rpv1*/*Run1* was less inherited (between 36% and 42% of individuals) in the populations 20_38 (‘Glera’ × ’Vc531.039’), 20_41 (‘Sk-00-1/7’ × ‘Glera’), 20_42 (‘Glera’ × ‘Vc156.1017’), and 20_43 (‘Glera’ × ‘Vc109.033’); that is, four of the eight populations segregating for the locus. Distortion occurred using both the resistance donor as the seed parent or the pollinating parent. Considering the *Rpv1*/*Run1* inheritance from Vc109.033, the distortion was high in the cross 20_43 with ‘Glera’, and the loci were inherited by 42% of progenies, while in the cross 21_45 with ‘Souvignier gris’, the distortion was not clear, and the loci were recorded in 45% of the seedlings.

*Rpv12* distortion was more intriguing. A higher inheritance of the locus was observed in the populations 19_33 (‘Glera’ × ’01-01-686’), 19_34 (‘Glera’ × ’01-01-881’), 20_43 (‘Glera’ × ‘Vc109.033’), and 21_45 (‘Souvignier gris’ × ‘Vc109.033’); that is, four of the seven populations segregating for the locus. In populations 19_33 and 19_34, the locus was inherited by about 60% of plants, while, in the populations 20_43 and 21_45, the locus was inherited by about 90% of the plants, suggesting a constant genotype-dependent impact of ‘Vc109.033’ (used as pollen donor) on the locus segregation.

Allelic data did not show specific limitations to resistance loci combinations inheritance. The 9 cross combinations performed provided a total of 42 combinations, 31 with at least one locus of resistance to *P. viticola* and one to *E. necator* (4974 plants—61% of the total), of which 8 with pyramided loci against both pathogens (1015 plants—13% of the plants) (Figure 3; Appendix A).

## 3. Discussion

### 3.1. Crude DNA Based MAS

The use of SSR markers in plant research is well documented. Simple and reliable approaches in SSR analysis are necessary to improve genotyping throughput and reduce costs in plant breeding. This is specifically important in MAS in which thousands of plants are analyzed, most seedlings are usually not retained, and, therefore, complete DNA extraction is not so essential. In grapevine research, direct PCR was implemented on single samples and different grapevine tissues for a variety fingerprinting [35]. In our study, we present a protocol for crude DNA extraction on 96 well plates and multiplex PCR that achieved a high success rate in MAS (98%), with a PCR specificity and data yield comparable to methods based on purified DNA, high throughput (thousands of samples per week), limited time of analysis (between 2.5 and 3.2 min per sample) and reduced costs (between 0.42 and 1.42 € per sample).

For crude DNA-based PCR, no or little adjustments to primer concentrations were performed in comparison to previous conditions for multiplex PCR on high-quality DNA. Therefore, the bottleneck associated with new time-consuming optimizations was avoided; for instance, in the concentration of each locus-specific primer pair and multiplex combinations, meeting a direct application of non-purified DNA in PCR. However, in a complete system, the optimization of well-balanced multiplex PCR based on the dynamics of the primers must be considered as a primary phase in new investigations [35].

Our protocol for crude DNA extraction and PCR may support molecular analysis in other plant species (Cipriani personal communication; [36]) without the need for further specific optimization except for the sampling. The protocol of the study used homemade reagents for obtaining crude DNA and preliminary information showed that it can be applied on other plant tissues (e.g., grapevine wood and rachis) and that crude DNA seemed suitable for the analysis with other Taq polymerases, either for direct or high specificity (HS) amplifications, without the need for an enhancer to increase the specificity and yield of PCR. However, young leaves represent the best solution for MAS application because they are easier and faster to sample and only low-volume applications of suggested Taq polymerases (final PCR volume between 12.5 and 20 µL) allowed a cost-effective application for MAS purposes which would be limited by the utilization of larger volumes (e.g., 50 µL). Finally, these observations suggest great flexibility for the applications of the proposed protocol because it does not rely on specific commercial reagents and PCR can be performed under different conditions.

The analysis of crude DNA provided reproducible and replicable results for rapid application for breeding purposes. In comparison to direct-PCR methods, it was not destructive and allowed multiple PCR reactions to be performed [5], and limited the issues and variable results associated with the amount and quality of plant material [4]. Our protocols also represent an easier and quicker solution in comparison to 96 plates extraction kits reutilization that showed a longer preparation and regeneration time than the application of the original ones [1]. However, we did not test the crude DNA in long-term conservation (more than 4 months), which can be one of the purposes of complete DNA purification.

In conclusion, the proposed protocol meets all MAS requirements providing several advantages such as sustainable costs, rapidity, flexibility, and reliability, which includes high yield and uniform amplification within multiplexed reactions. Moreover, there is a good potential to test the protocol in different species and plant tissues, and for its amelioration (increasing the informativeness and reducing costs) by improving the degree of multiplexing, trying alternative materials (e.g., Collection Microtubes) and reagents (e.g., Taq polymerase), and reducing the analysis time (e.g., optimizing the times for crude DNA obtaining). Future aims would be to test our crude DNA with other molecular analyses that could further improve grape breeding efficacy. SNP analysis from crude DNA extract was already proposed in the molecular breeding of different crops [3]. For grapevine, high-throughput strategies to follow genetic variants haplotypes were proposed and applied to date on purified DNA [37].

### 3.2. Grapevine Resistance Locigenotyping and Segregation

Distorted segregations in grapevine were sometimes reported in genetic mapping studies. Distortions occurred in different populations and genomic regions, for instance in chr 11 and 13 of *Vitis rotundifolia* descendants [38], or in chr 13 [39] and 14 [40] in *V. vinifera*, but they were always independent of resistance loci regions. Our analysis showed distorted segregations for *Rpv1*/*Run1*, derived from *V. rotundifolia* [20,26], and located on chr 12, and for *Rpv12*, derived from *V. amurensis* [25], located on chr 14. Distorted segregations may be unpredictable and occur because of post-zygotic lethal combinations that influence the viability of zygotes, germination of seeds, and seedlings’ survival [38]. In the present study, *Rpv1*/*Run1* was less inherited by several populations. Similarly, Sanchez-Mora et al. [41] reported a low number of progenies homozygous for the locus. *Rpv1*/*Run1*-carrying parents were either the seed parent or the pollen donor confuting the possibility of a mechanism affecting the pollen grains or ovules as responsible for the distortion. For *Rpv12*, we observed that in some populations, a higher percentage of plants (from 60 to 90%) inherited the locus. The cross utilization of the resistant donor VC109.33 with multiple seed parents showed that the distortions were not cross-specific and may be associated with a gametic selection affecting pollen grains. However, such a hypothesis should be verified by using VC109.33 as the seed parent in new crosses. The described segregation distortions provided good indications for the suitability of crossed accession as resistant donors. Selections providing an increased inheritability of certain loci may represent an alternative solution to “Locus specific homozygous (LSH) lines” (homozygous for selected resistance regions) [42] to increase the percentage of resistant progenies. This is particularly necessary in a pyramiding process in which the number of desired plants decreases with the aim of staking more loci. Lastly, despite the unpredictable segregation patterns, the present set of cross combinations provided an affordable number of plants with pyramided loci against *P. viticola* and/or *E. necator.* In general, pyramiding is a breeding strategy to increase the level, stability, and durability of plants’ resistance to pathogens [43,44].

The preliminary characterization of resistance loci in grapevine seedlings provided a good indication of the possible plant defense response and resistance level [33,45]. The literature provides good phenotyping examples for single *Rpv*, *Ren*, and *Run*, while only some combinations of pyramided loci were described [41,45,46]. In our breeding program, we introgressed in breeding lines the locus *Ren1.2* and generated both *Rpv* and *Ren*/*Run* combinations not described to date (e.g., in populations 21_44 and 21_45). We will focus our selection on plants carrying pyramided loci to both pathogens because they represent an interesting resource to identify new varieties, use as resistance donors, and carry out new phenotypic studies on *Rpv*/*Ren*/*Run*-mediated resistances.

## 4. Materials and Methods

### 4.1. Plant Material

Several controlled crosses between grapevine accessions were performed at CREA—Research Centre for Viticulture and Enology (CREA-VE), Susegana, Treviso, Italy (45°51′07.6″ N, 12°15′28.6″ E) between 2019 and 2021. In 2019 and 2020, ‘Glera’, an Italian *V. vinifera* variety susceptible to *P. viticola* and *E. necator*, was crossed with the resistant breeding selections ‘01-01-686’, ‘01-01-881’, ‘VC109.033’ (or ‘Pinot Iskra’), ‘VC156.1017’, ‘VC531.039’, ‘SK 00-1/7’, and ‘Floreal’, carrying, in different combinations, the resistance loci to *P. viticola Rpv1* [20], *Rpv3.1* [21], and *Rpv12* [25], and the resistance loci to *E. necator Run1* [26], *Ren1.1* [27], *Ren3*, and *Ren9* [28,29]. In 2021, the CREA-VE breeding selection ‘14_05d_023’ (‘Glera’ × ‘Solaris’), carrying loci *Rpv3.3* [23], *Rpv10* [24], *Ren3*, and *Ren9*; the resistant variety ‘Souvigner gris’, carrying *Rpv3.2* [22], *Ren3*, and *Ren9*, and ‘Shavtsitska’, carrying *Ren1.2* [39], were also used in new crosses for resistance-loci-pyramiding purposes. Details on the cross combinations and information on parental plants are reported in Appendix A. Inter-institutional agreements permitted the plant material and pollen from different germplasm repositories to be shared and to produce the seeds used in the present research.

The seeds of the population were conserved as described in De Nardi et al. [33] and sown in spring in 170 wells plateau in blond peat moss on a heated surface. Germinated plants were re-potted in 2 L pots in a mixture of different peats, then kept in a greenhouse under natural light and a temperature < 29 °C, fed with a nutritive solution, and sprayed every two weeks for pests and diseases just to guarantee the presence of young apical leaves for sampling.

### 4.2. Genotyping of Parental Plants and Cross Populations

For each cross parent, total DNA was extracted with ‘DNeasy Plant Mini Kit’ (Qiagen, Hilden, Germany), and the SSRs, Gf09-47 [24], Gf15-28, Gf15-30 [22], UDV350, UDV360 [25], UDV734 [47], VMC4f3.1, VMC8g9 [48], VMC7f2 [49], SC8-0071-014, and sc47_20 [50] were used to screen the alleles of the *Rpv*, *Ren*, and *Run*, as described in De Nardi et al. [33].

For each germinated seedling, non-purified DNA (crude) was obtained from a single leaf sample with home-made reagents and the custom protocol described below:About 50 mg of a young and shiny leaf were collected in 96-tube (1.2 mL) Collection Microtubes plates (Qiagen, Hilden, Germany);Samples were lyophilized (Christ Alpha 2–4 LD, Martin Christ Gefriertrocknungsanlagen GmbH, Osterode am Harz, Germany) for 24–48 h, sealed, and stored at room temperature until the processing date (for up to one month);One stainless steel bead of 3.2 mm was added to each tube, and the samples were processed twice at 30 Hz for 120 s (by inverting the internal/external side of the plates) in a Tissue-Lyser II instrument (Qiagen, Hilden, Germany) in order to ground the tissues to a fine powder;A total of 450 μL of home-made lysis buffer (Tris HCl 200 mM—pH 8, NaCl 250 mM, EDTA 25 mM and SDS 0.5%; (Lemke et al. [1] with minor modifications), pre-heated at 65 °C, was added to each sample. Then, the plates were mixed at 18 Hz for 50 s in the Tissue Lyser II, incubated in an oven at 65 °C for 15 min, and shaken 2–3 times during the lysis;A total of 130 μL of home-made precipitation buffer (KOAC 5 M—pH 6,5; [1]) was added to each sample, the plates were manually mixed for 15 s, and then stored in ice for 10 min;Collection Microtubes plates were centrifuged at 5700× *g* for 15 min to separate the solid particles from the solution;A total of 100 µL of the extraction solution of each sample was moved to the 96-well (200 µL) PCR plates;The PCR Plates were centrifuged at 5700× *g* for 15 min again to separate the solid particles transferred in the previous step;An aliquot of the extraction solutions diluted 20-fold was prepared and stored at −20 °C as ‘crude DNA’ in new 96-well PCR plates.

A scheme of the protocol for crude DNA extraction is represented in Figure 4.

MAS of seedlings was performed by using subsets of SSR flanking the *Rpv*, *Ren*, and *Run* analyzed in the cross parents according to the inheritable resistance sources (Table 1 and Appendix A). Multiplex PCR reactions (one duplex, two triplex, and four four-plex) were performed in 20 μL volume containing 1 × MyTaq Plant PCR Mix (MyTaq Plant PCR Kit, Meridian Life Science, Memphis, TN, USA), from 0.05 to 0.29 μM of each primer, 1 μL of crude DNA and water for molecular biology. The PCRs were performed under the following thermal profile: 94 °C for 4 min, followed by 35 cycles at 94 °C for 45 s, 55 °C for 1 min and 30 s, 72 °C for 1 min, and final elongation of 10 min at 72 °C in a T-100 Thermal Cycler (Bio-Rad, Hercules, CA, USA). A lower volume of 12.5 μL was used for duplex PCR. A higher volume of 25 μL, 1.5-fold increased primer concentrations, and up to 42 PCR cycles were implemented for problematic samples. Forward SSR primers were 5′-end-labeled with different fluorescent dyes (Appendix A): 6-FAM, HEX, NED, PET, and VIC (Thermo Fisher Scientific, Waltham, MA, USA).

The sequencing reaction was prepared to a final volume of 10 μL as follows: 0.50 μL PCR product/products (whenever possible, the amplicons were pooled in 1:1 proportion and analyzed simultaneously), 0.16 μL of Gene Scan 500 LIZ (Thermo Fisher Scientific, Waltham, MA, USA), and 9.34 μL of Hi-Di Formamide (Thermo Fisher Scientific, Waltham, MA, USA). Capillary electrophoresis was carried out using an ABI Prism 3130 × l DNA analyzer (Thermo Fisher Scientific, Waltham, MA, USA) with the polymer POP-7 (Thermo Fisher Scientific, Waltham, MA, USA), and allelic data were obtained with GeneMapper 4.0 software (Thermo Fisher Scientific, Waltham, MA, USA).

## 5. Conclusions

The crude DNA-based multiplex PCR of SSRs provides several technical advantages for the fingerprinting of plant accessions and MAS of progenies. These advantages rely on considerable costs and labor savings and on an easy, flexible, and transferable protocol for different plant species.

The identification of distorted segregations for grapevine resistance loci provides useful information for breeders, alerting to the presence of unlucky cross combinations with lower inheritance of the loci, and on the opportunity to use parental plants able to transfer the desired loci to most of the offspring. Progenies carrying pyramided resistance represent valuable new plant material for original studies and for the breeding program.

## Figures and Tables

**Figure 1 plants-13-02001-f001:**
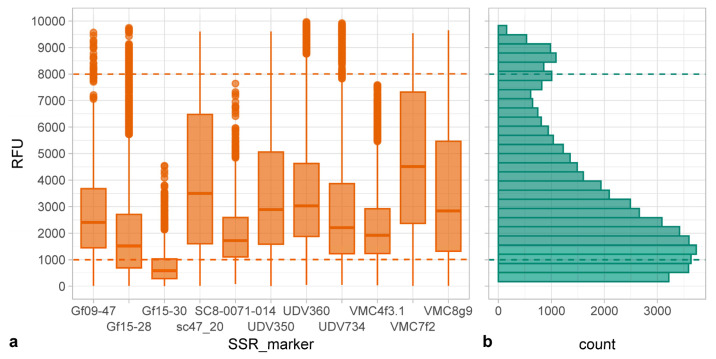
(**a**) Relative fluorescence units (RFU) recorded in capillary electrophoresis for the simple sequence repeats markers (SSR) for the marker-assisted selection. Dashed lines delimited the optimal RFU (1000–8000) for semi-automated analysis with GeneMapper 4.0 software. (**b**) Histogram summarizing the distribution of all the RFU yielded (*x*-axis represents the counts for each bin displayed).

**Figure 2 plants-13-02001-f002:**
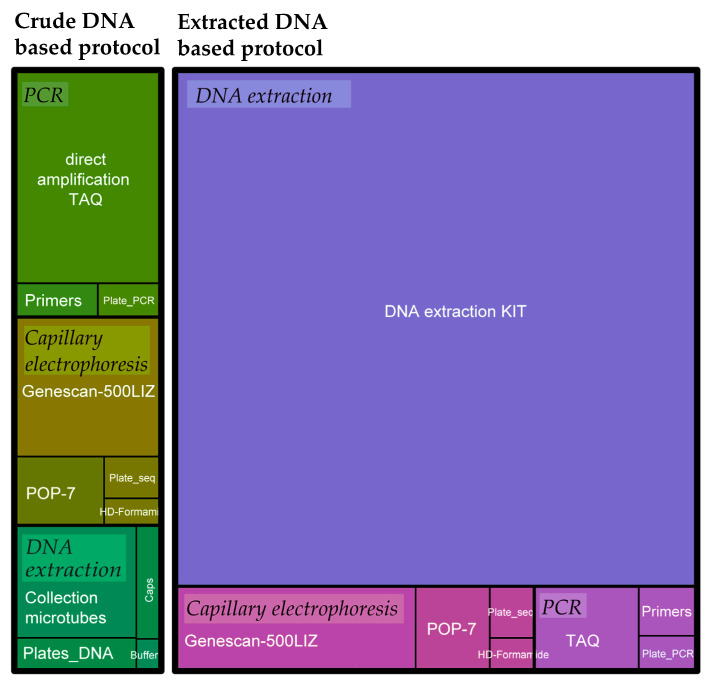
Proportional representation of costs per single plant sample for the optimized MAS protocol based on non-purified DNA (crude) amplification (green) and on traditional DNA extraction Kit (Plant DNeasy Mini Kit, Qiagen, Hilden, Germany), and amplification (violet). In the crude DNA-based protocol, costs are reduced by at least 3.5 times: costs associated with PCR increase for the utilization of a Taq polymerase adapted for direct amplifications, while reagents and material for DNA obtaining represent the cheapest part of the process.

**Figure 3 plants-13-02001-f003:**
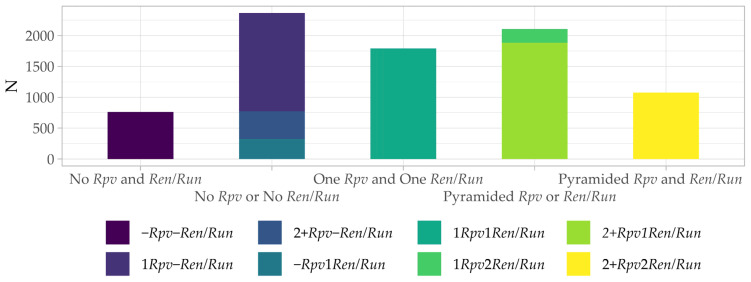
Number of seedlings (N) that inherited none (-), one (1), and two (2) or more than two (2+) *Rpv*, *Ren*, and *Run* loci. Details for the number of seedlings carrying the different combinations of resistance loci are reported in Appendix A.

**Figure 4 plants-13-02001-f004:**
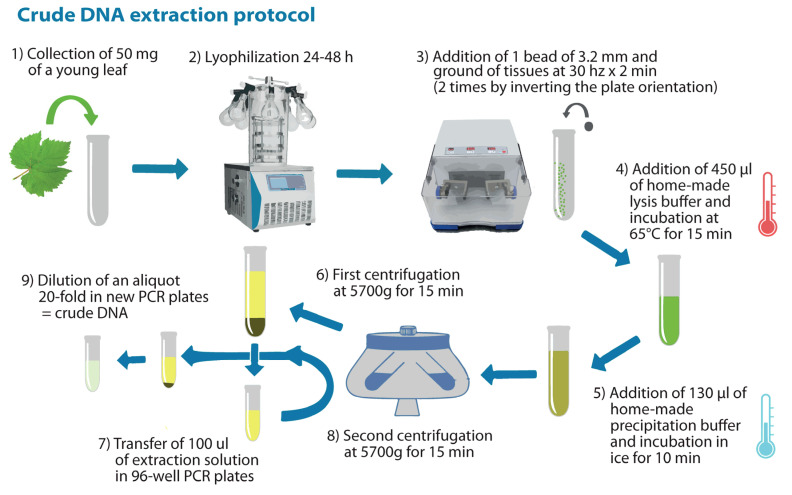
Schematic representation of the protocol implemented to obtain the crude DNA for the amplification of simple sequence repeats (SSRs) markers for the marker-assisted selection (MAS).

**Table 1 plants-13-02001-t001:** Resistance loci to *P. viticola* (*Rpv*) and *E. necator* (*Run* or *Ren*) segregating in the population of the study, simple sequence repeats markers (SSRs) utilized for the marker-assisted selection, and alleles associated with the resistance. ‘/’ indicates loci located close on the same chromosome in the grape reference genome ‘PN40024’ and follow with the same SSR.

Pathogen	Resistance Locus	Chromosome	Locus-Associated SSR	Resistance-Associated Alleles ^6^
*P. viticola*	*Rpv3.1* *Rpv3.2* *Rpv3.3*	18	UDV734 VMC7f2	233 ^3^–237 ^1^–241 ^2^ 200 ^2;3^–211 ^1^
	*Rpv10*	9	Gf09-47	297
	*Rpv12*	14	UDV350 UDV360	308
				208
*P. viticola*/ *E. necator*	*Rpv1*/*Run1*	12	VMC4f3.1	192
			VMC8g9	157
*E. necator*	*Ren1.1* *Ren1.2*	13	SC8-0071-014 sc47_20	147 ^4^–149 ^5^ 206 ^4^–208 ^5^
	*Ren3*/*Ren9*	15	Gf15-30	445
			Gf15-28	342

^1^ *Rpv3.1* alleles; ^2^ *Rpv3.2* alleles; ^3^ *Rpv3.3* alleles; ^4^ *Ren1.1* alleles; ^5^ *Ren1.2* alleles; ^6^ Data in our analysis.

**Table 2 plants-13-02001-t002:** Distorted segregations identified for the resistance loci segregating in the cross-population of the study with the expected and observed percentages of inheritance.

Pop. Code	Seed Parent	Pollen Donor	Segregating Loci	Segregation Distortions		
				Locus	Expected %	Observed %
19_33	Glera	01-01-686	*Rpv12*; *Rpv1*/*Run1*	*Rpv12*	0.50	0.59
19_34	Glera	01-01-881	*Rpv12*; *Rpv1*/*Run1*	*Rpv12*	0.50	0.65
19_35	Glera	Floreal	*Rpv3.1*; *Rpv1*/*Run1*; *Ren3*/*Ren9*	None		
20_38	Glera	Vc531_039	*Rpv3.1*; *Rpv12*; *Rpv1*/*Run1*; *Ren1.1*	*Rpv1*/*Run1*	0.50	0.40
20_41	SK-00-1/7	Glera	*Rpv3.1*; *Rpv12*; *Rpv1*/*Run1*; *Ren3*/*Ren9*	*Rpv1*/*Run1*	0.50	0.36
20_42	Glera	Vc156_1017	*Rpv12*; *Rpv1*/*Run1*	*Rpv1*/*Run1*	0.50	0.42
20_43	Glera	Vc109_033	*Rpv12*; *Rpv1*/*Run1*; *Ren3*/*Ren9*	*Rpv12*	0.50	0.90
				*Rpv1*/*Run1*	0.50	0.40
21_44	Shavtsitska	14_05d_023	*Rpv3.3*; *Rpv10*; *Ren1.2*; *Ren3*/*Ren9*	None		
21_45	Souvignier gris	Vc109_033	*Rpv3.2*; *Rpv12*; *Rpv1*/*Run1*; *Ren3*/*Ren9*	*Rpv12*	0.50	0.87
				Possible distortions for *Rpv1*/*Run1* and *Rpv3.2*		

## Data Availability

The authors confirm that the data supporting the findings of this study are available within the article [and/or] its Appendix A.

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
