# Peer review of "A Sustainable Strategy for Marker-Assisted Selection (MAS) Applied in Grapevine (Vitis spp.) Breeding for Resistance to Downy (Plasmopara Viticola) and Powdery (Erysiphe Necator) Mildews"

_plants, 2024, doi:10.3390/plants13142001_

Round 1

Reviewer 1 Report

Comments and Suggestions for Authors

This paper describes a rapid and inexpensive method of DNA extraction for the analysis of large numbers of samples to study the genetic characteristics of plants. Therefore, the topic of this manuscript (Ms) is relevant to Plants. However, there are several important remarks to the Ms. The Ms needs major revisions.

1) This manuscript (Ms) contains misprints, mistakes in English grammar and in the writing style. I recommend that the authors should use some help of a native English speaker or send the Ms to an English Editing Service that proofreads scientific writing. It is very difficult to understand different parts of the manuscript in its present form.

 2) The authors should significantly revise the text, because the Ms contains many specific terms and abbreviations, which worsens the perception of the Ms text, e.g.:

a) improve explanation of Table 1, e.g. Chr, SSR markers, Resistance alleles, etc.

b) improve explanation of Fig. 1, e.g. a) and b) parts, count, Gf09-47, Gf15-28, etc.

c) improve explanation of Fig. 3, e.g. N, whose seedling is it, etc.

d) can authors explain the theory of using primers in the form of a figure?

 3) The authors should carefully double-check the presented DNA extraction procedure and clarify and explain some points:

a) Line 317: what is the total well capacity of wells in the used 96-well plates?

b) Line 332: “100 µl of clean surnatant” – surnatant of supernatant?

c) Line 333: what is the 7th step for?

d) Can authors present the described method in the form of a figure?

 4) Line 307: authors should explain growth conditions in greenhouse (e.g. temperature, illumination, day/night duration, etc.).

 Minor:

5) Line 38: “Microsatellites, or simple sequence repeats” correct to “Microsatellites or simple sequence repeats”.

6) Line 42: “depended on human work” correct to “depended on manual labor”.

7) Line 88, 123, 126, 127, 130, 134, Fig. 2, 223, 226, 245: “TAQ polymerase” correct to “Taq polymerase”.

8) Fig. 2: include the names of the used DNA extraction kits.

9) Line 341: explain “rDNA”.

10) Line 343: “1m,” correct to “1 min,”.

Comments on the Quality of English Language

This manuscript (Ms) contains misprints, mistakes in English grammar and in the writing style. I recommend that the authors should use some help of a native English speaker or send the Ms to an English Editing Service that proofreads scientific writing.It is very difficult to understand different parts of the manuscript in its present form.

Author Response

Manuscript review - Round 1

Open Review – Reviewer 1

  • Quality of English Language

( ) I am not qualified to assess the quality of English in this paper

( ) English very difficult to understand/incomprehensible

( ) Extensive editing of English language required

(x) Moderate editing of English language required

( ) Minor editing of English language required

( ) English language fine. No issues detected

  • Does the introduction provide sufficient background and include all relevant references?

Yes         Can be improved              Must be improved           Not applicable

(x)                          ( )                                         ( )                                         ( )

  • Is the research design appropriate?

Yes         Can be improved              Must be improved           Not applicable

( )                          ( )                                         (x)                                         ( )

  • Are the methods adequately described?

Yes         Can be improved              Must be improved           Not applicable

( )                          ( )                                         (x)                                         ( )

  • Are the results clearly presented?

Yes         Can be improved              Must be improved           Not applicable

( )                          ( )                                         (x)                                         ( )

  • Are the conclusions supported by the results?

Yes         Can be improved              Must be improved           Not applicable

( )                          (x)                                         ( )                                         ( )

Comments and Suggestions for Authors

This paper describes a rapid and inexpensive method of DNA extraction for the analysis of large numbers of samples to study the genetic characteristics of plants. Therefore, the topic of this manuscript (Ms) is relevant to Plants. However, there are several important remarks to the Ms. The Ms needs major revisions.

1) This manuscript (Ms) contains misprints, mistakes in English grammar and in the writing style. I recommend that the authors should use some help of a native English speaker or send the Ms to an English Editing Service that proofreads scientific writing. It is very difficult to understand different parts of the manuscript in its present form.

Answer: After the editing according to reviewers comments, we submitted our manuscript and the supplementary materials to an English editing service provider. All changes about English grammar are reported in red color. We believe that it has improved now.

 2) The authors should significantly revise the text, because the Ms contains many specific terms and abbreviations, which worsens the perception of the Ms text, e.g.:

Answer: The authors entirely revised the text adding details that can improve the understanding of specific terms and abbreviation. All the suggestions reported by the reviewer were addressed.

  1. a) improve explanation of Table 1, e.g. Chr, SSR markers, Resistance alleles, etc.

Answer: Explanation and header of Table 1 was extended.

  1. b) improve explanation of Fig. 1, e.g. a) and b) parts, count, Gf09-47, Gf15-28, etc.

Answer: Caption of Figure 1 was extended.

  1. c) improve explanation of Fig. 3, e.g. N, whose seedling is it, etc.

Answer: Figure 3 was modified to improve the understanding of axis labels and legend. The figure would like to summarize the information contained in Tables S4. In the caption was reported that in this supplementary material the information on resistance combinations can be found.

  1. d) can authors explain the theory of using primers in the form of a figure?

Answer: primers utilization to yield specific PCR products is a consolidate practice in molecular biology. We believe that is not necessary to dedicate on figure for this topic.

 3) The authors should carefully double-check the presented DNA extraction procedure and clarify and explain some points:

Answer: Unfortunately, any sample was process twice to directly compared the results and once optimized the protocol we did not buy new extraction kit. The consistency of the method and of the results was basically verified by the early characterization of resistance alleles in parentals plants on extracted DNA. Following, in MAS only parental alleles were expected and effectively yielded. Furthermore, MAS by using the same markers and capillary electrophoresis method is implemented in the laboratory since long time as stated in the introduction. We added some sentences in the text (line 99-103) to state the identification in the MAS of the allele/PCR amplicons yielded in parental plants and the similarity of the new analysis results with the former analysis implemented in previous paper (De Nardi et al., 2019) in which standard DNA extraction protocol was utilized.

  1. a) Line 317: what is the total well capacity of wells in the used 96-well plates?

Answer: The total Collection Microtubes capacity is 1.2 ml, the information was added.

  1. b) Line 332: “100 µl of clean surnatant” – surnatant of supernatant?

Answer: We referred to the solution we modify the term with ‘extraction solution’.

  1. c) Line 333: what is the 7th step for?

Answer: Step 7th was done to separate eventual solid particles transferred from the previous 1.2 ml tube. The information was added.

  1. d) Can authors present the described method in the form of a figure?

Answer: An image to schematically present the protocol used for crude DNA extraction was prepared and added.

 4) Line 307: authors should explain growth conditions in greenhouse (e.g. temperature, illumination, day/night duration, etc.).

Answer: Essential details on growth conditions were added. We don’t consider essentials further details because leaves were necessary only to obtain crude DNA for the marker assisted selection.

 Minor:

5) Line 38: “Microsatellites, or simple sequence repeats” correct to “Microsatellites or simple sequence repeats”.

Answer: Done.

6) Line 42: “depended on human work” correct to “depended on manual labor”.

Answer: Done.

7) Line 88, 123, 126, 127, 130, 134, Fig. 2, 223, 226, 245: “TAQ polymerase” correct to “Taq polymerase”.

Answer: Done.

8) Fig. 2: include the names of the used DNA extraction kits.

Answer: We include the DNA extraction kit used as reference.

9) Line 341: explain “rDNA”.

Answer: we correct in 1 μl crude DNA.

10) Line 343: “1m,” correct to “1 min,”.

Answer: Done.

Reviewer 2 Report

Comments and Suggestions for Authors

The manuscript describes a protocol of marker-assisted selection (MAS) for grapevine resistance breeding, and the protocol and results presented are useful tools for grape breeders. Overall, the ms is well organized. However, before to be accepted for published, some comments need to be addressed:

1) In line 138-139: the authors caculated the time for  lyophilization management, crude DNA extraction is about 150-180 minutes per 192 samples. However,  lyophilization management is time-consuming job, in line 318, the ms describes it take 24-48 h. In this case, is it sitll high efficiency? Do we need caculate the cost of the electricity charge for lyophilization management?

2) Line 318:: please provide the manufacturer and model of equipment for lyophilization.

3) resistance phenotype analysis of random selection seedlings need to be provided to confirm the MAS results.

4) line 29: What does the “species" mean?

Author Response

Manuscript review - Round 1

Open Review – Reviewer 2

  • Quality of English Language

(x) I am not qualified to assess the quality of English in this paper

( ) English very difficult to understand/incomprehensible

( ) Extensive editing of English language required

( ) Moderate editing of English language required

( ) Minor editing of English language required

( ) English language fine. No issues detected

  • Does the introduction provide sufficient background and include all relevant references?

Yes         Can be improved              Must be improved           Not applicable

(x)                          ( )                                         ( )                                         ( )

  • Is the research design appropriate?

Yes         Can be improved              Must be improved           Not applicable

( )                          (x)                                         ( )                                         ( )

  • Are the methods adequately described?

Yes         Can be improved              Must be improved           Not applicable

(x)                          ( )                                         ( )                                         ( )

  • Are the results clearly presented?

Yes         Can be improved              Must be improved           Not applicable

(x)                          ( )                                         ( )                                         ( )

  • Are the conclusions supported by the results?

Yes         Can be improved              Must be improved           Not applicable

( )                          (x)                                         ( )                                         ( )

Comments and Suggestions for Authors

The manuscript describes a protocol of marker-assisted selection (MAS) for grapevine resistance breeding, and the protocol and results presented are useful tools for grape breeders. Overall, the MS is well organized. However, before to be accepted for published, some comments need to be addressed:

1) In line 138-139: the authors calculated the time for lyophilization management, crude DNA extraction is about 150-180 minutes per 192 samples. However, lyophilization management is time-consuming job, in line 318, the MS describes it take 24-48 h. In this case, is it still high efficiency? Do we need calculate the cost of the electricity charge for lyophilization management?

Answer: We focused the research work on consumables costs (i.e., the DNA extraction kit) and we provided indication about the physical time (presence of one operator) necessary to process a sample in our lab conditions. The freeze-drying time is not counted because the instrument work in autonomy without occupies one operator during the lyophilization, alternatively we should consider also the PCR program (2.5 h) and the capillary electrophoresis runs (35-40 minutes each 16 samples). Furthermore, on these phases the throughput could be improved by using bigger lyophilizers, 384-well block PCR machine and 96-capillary sequencers. In costs comparison, the impact of electricity is not so important because the instrument required and the time of utilization are the same of those to obtain extracted DNA. Lastly, the lyophilization, as in standard DNA extraction methods, could be also substituted by the utilization of liquid nitrogen if desired. In line 249-250 we stated that the time of the analysis could be improved.

2) Line 318: please provide the manufacturer and model of equipment for lyophilization.

Answer: Done.

3) resistance phenotype analysis of random selection seedlings needs to be provided to confirm the MAS results.

Answer: Phenotyping bioassays were performed on selection seedlings in the past to validate the MAS and it is stated in the discussion in lines 283-284. The article aims to propose a protocol for MAS and for the species of the study association between markers, alleles and resistance trait are already consolidated. 

4) line 29: What does the “species" mean?

Answer: In this contest, the term “species” is referred to the different varieties of plants that can be mainly classified based on certain features, including growth habits, in 5 main types of plants: herbs, shrubs, trees, creepers, and climbers.

Round 2

Reviewer 1 Report

Comments and Suggestions for Authors

The authors have addressed the comments and improved the manuscript. I think that the manuscript can be accepted.

Comments on the Quality of English Language

Minor editing is required.

Reviewer 2 Report

Comments and Suggestions for Authors

The manuscript is ok now